# Primary productivity measurements in the Ross Sea, Antarctica: A regional synthesis

Walker O. Smith, Jr.[1,2]

[1] School of Oceanography, Shanghai Jiao Tong University, Shanghai, 200300, PRC

[2] Virginia Institute of Marine Science, William & Mary, Gloucester Pt., VA, 23062, USA

*Correspondence to*: Walker O. Smith, Jr. (wos@vims.edu)

**Abstract.** Polar systems are undersampled due to the difficulty of sampling remote and challenging environments; however, these systems are critical components of global biogeochemical cycles. Measurements on primary productivity in specific areas can quantify the input of organic matter to food webs, and so are of critical ecological importance as well. However,

long-term measurements using the same methodology are available only for a few polar systems. Primary productivity measurements using $^{14}$C-uptake incubations from the Ross Sea, Antarctica, are synthesized, along with chlorophyll concentrations at the same depths and locations. A total of 19 independent cruises were completed, and 449 stations occupied where measurements of primary productivity (each with 7 depths) were completed. The incubations used the same basic simulated *in situ* methodology for all. Integrated water column productivity for all stations averaged $1.10 \pm 1.20$ g C m$^{-2}$ d$^{-1}$,

and the maximum was 13.1 g C m$^{-2}$ d$^{-1}$. Annual productivity calculated from the means throughout the growing season equalled 146 g C m$^{-2}$ yr$^{-1}$. The mean chlorophyll concentration in the euphotic zone (the 1% irradiance level) was $2.85 \pm 2.68$ mg m$^{-3}$ (maximum observed concentration was 19.1 mg m$^{-3}$). Maximum photosynthetic rates above the 30% isolume (normalized to chlorophyll) averaged $0.98 \pm 0.71$ mg C (mg chl)$^{-1}$ h$^{-1}$, similar to the maximum rate found in photosynthesis-irradiance measurements. Productivity measurements are consistent with the temporal patterns of biomass found previously, with biomass

and productivity peaking in late December; mixed layers were at a minimum at this time as well. Estimates of plankton composition also suggest that pre-January productivity was largely driven by the haptophyte *Phaeocystis antarctica*, and summer productivity by diatoms. The data set will be useful for a comparison to other Antarctic regions and provide a basis for refined bio-optical models of regional primary productivity and biogeochemical models for the Southern Ocean.

## 1        Introduction

A quantitative assessment of the ocean's primary productivity (the rate at which carbon dioxide is reduced to organic matter by marine phytoplankton photosynthesis) is a critical variable in understanding the ecology and biogeochemistry in marine systems. Phytoplankton, being the base of marine food webs, grow at different rates in the ocean, rates that are regulated by temperature, nutrients, and irradiance; they are in turn grazed by herbivorous organisms that both incorporate the autotrophic organic matter into their own tissues, as well as recycle it by oxidizing a portion for energy use and release

inorganic materials back into the ocean. Phytoplankton are also critical to marine elemental cycles, as they are central in cycling carbon, nitrogen, phosphorus, iron, and all bio-active elements. In the carbon cycle they not only fix $CO_2$ into organic matter, but release dissolved organic matter that is used by heterotrophic microbes. They also can sink and be incorporated into larger particles, and the flux of organic material to depth is the essential driver of what is known as the biological pump (Basu and Mackey, 2018). The processes that contribute to the vertical movement of organic matter can sequester carbon from the atmosphere for periods from years to centuries.

Primary productivity in the ocean traditionally has been measured by collecting water, adding a radioactive tracer, and incubating samples in the irradiance environment from which the sample was taken, and quantifying the radioactive uptake after incubation. This procedure was introduced by Steemann Nielsen (1952) and has been applied to nearly every ocean numerous times. However, the ocean is far too large to measure productivity synoptically, and as a result numerous bio-optical models have been derived to estimate primary productivity using remotely sensed information (temperature, chlorophyll concentrations, irradiance; e.g., Behrenfeld and Falkowski, 1997a,b). These models have enabled oceanographers to estimate productivity on large space and time scales, and have also been used in more restricted analyses of productivity (Mouw and Yoder, 2005; Smith et al., 2000, 2021; Oliver et al., 2021). Two critical aspects of the original Behrenfeld and Falkowski (1997a,b) models were both based on results from primary productivity measurements that used [14]C-uptake – the relationship between the maximum rate of photosynthesis as a function of temperature (then fit to a 7th-order polynomial) and the photoinhibition estimate.

Carbon radioisotope measurements of primary productivity are extremely sensitive, and other methods (e.g., oxygen changes) cannot discriminate the small changes that characterize many marine systems. As a result, a comparison among different procedures has been difficult. Numerous questions about the interpretation of [14]C-data also have been posed. For example, concerns were expressed with regard to whether the method measured net or gross production, the effects of light and dark respiration, and the impact of heterotrophic plankton (Marra, 2009). While these uncertainties have been repeatedly acknowledged, it appears that long (e.g., 24-h) measurements approximate net production, but this has not been rigorously examined in polar systems (Marra and Barber, 2004; Marra, 2009). However, it is likely that the measurements reported herein should be interpreted as net production.

Methods measuring [14]C-uptake vary among different investigators. Some studies used *in situ* incubations where samples were returned to the ocean at the depths from which they were taken, thus insuring that the same irradiance environment (Marra et al., 2021), while others use neutral density screens and on-deck incubators that are cooled by running seawater (e.g., Barber et al., 1997). Some studies use blue filters to correct for changes in spectral quality within the water column, while others do not. Older studies used small glass bottles, but it was subsequently determined that trace metals could bind to glass and potentially alter estimates or productivity. Bottle size also was shown to be important, as small bottles tended to result in the death of microzooplankton and disrupt the cycling of nitrogen (Eppley, 1982). Length of incubation is also variable, with some measurements being relatively short (6 h or less) while others encompass the entire 24-h photoperiod. Time of sampling during the day also varied, with some sampling at dawn (but using irradiance profiles from the previous day), where others

sampled whenever possible. All methods enclose samples in bottles and thus remove the plankton from the natural, turbulent environment. As a result, these differences create challenges when comparing productivity estimates.

Polar regions are even more challenging with regard to measurements of primary productivity. Cruises to remote regions are infrequent, and often completed during the summer when ice is reduced or absent, and storms less frequent. As such, temporal sampling is far from uniform. The environmental features of polar systems are also unusual. For example, low temperatures mean that growth rates tend to be slow, and 24-h incubations are often used. Irradiance durations are also diverse, as photoperiods in many polar settings during summer are 24 h (although local noon irradiance values are at least an order of magnitude greater than those at local midnight). Ice, when encountered, greatly reduces the *in situ* irradiance, and thus on-deck incubations may not truly represent the irradiance environment from which samples were taken.

This report summarizes rate measurements collected from a relatively small region in the Southern Ocean, the Ross Sea, Antarctica. While the Ross Sea covers a small portion of the entire Southern Ocean, it is considered to be the most important region for the removal of $CO_2$ from the atmosphere as a result of its large productivity (Arrigo et al., 2008). It also has been studied intensively since the first International Geophysical Year in 1958. The measurements synthesized here were done by a single investigator, using methods that were the largely the same over the span of ca. 25 years. They are not continuous through space or time, but represent a unique data set that should be of use to those interested in validating remote sensing-based productivity models and biogeochemical models of the region.

## 2 Data

Primary productivity was measured on 19 cruises in the Ross Sea, Antarctica (Table 1) from 1983 – 2006 using simulated *in situ* incubations (e.g., Smith et al., 2000). All stations (n = 499) included in this analysis were located on the continental shelf (Fig. 1). Stations were not selected for a geographically even distribution, but often were a function of ice and chlorophyll concentrations (as well as other factors). Thirty-nine stations were sampled from 1980-1989, 299 from 1990-1999, and 111 from 2000-2009; 11, 74, 92, 21, and 86 stations were sampled in October, November, December, January, and February, respectively. While interannual variability does occur (e.g., Smith et al., 2006, 2011), the seasonal variability is far greater than that observed among years (Smith et al., 2014). Samples were taken from known isolumes (determined usually by photosynthetically active radiation (PAR) sensors on the CTD-Niskin system, but in the 1980 cruises by use of a Secchi disk), inoculated with ca. 100 µCi $HCO_3^-$ in 5% KCl (pH 9.6), and incubated for 24 h. In the earliest cruises (1983) 125 mL glass bottles were used, but after 1990 polycarbonate bottles were adopted (280 mL). Either individual bottles were wrapped in neutral density screens, or incubators had tubes that were wrapped in neutral density screens and unwrapped bottles placed inside. After 1992 all incubators used blue filters (Cinemills Corp. #M144; the same filters that were used in Smith and Donaldson (2015) photosynthesis-irradiance measurements) as well at isolumes of 30% and below. During all cruises except NBP97-02, photoperiods were 24 h, and therefore samples were collected independent of local time; incubations began less than 20 minutes after sample collection. On all cruises additional phytoplankton variables were measured (e.g., chlorophyll, particulate organic carbon and nitrogen, biogenic silica, photosynthetic pigments, taxonomic composition), but only

chlorophyll concentrations, mixed layer depths, and the dominant phytoplankton group are included here. Routine oceanographic data (temperature, salinity, oxygen profiles) are also available for each cruise, as well as additional particulate material analyses.

Ice concentrations were variable during the cruises. During some stations ice cover was 100% (e.g., all stations during NBP97-02), and in others ice was absent. Most cruises included stations that were in variable ice cover as well as open water. Sampling in ice-covered waters is possible and routine, as long as the CTD can access water; however, determining the *in situ* irradiance is not direct using routine ship methodologies. Even in ice-covered waters, the CTD-derived PAR determinations of irradiance were used to sample. These estimates of isolumes likely overestimated the actual depth of isolumes, but the

degree of overestimation is unknown. For example, in waters with 10% ice cover, measured isolume depths are likely accurate, but in regions with 80% ice cover and more, isolume depths likely vary from the true depths (Smith, 1995).

     Incubations present additional challenges. For example, during periods when snowfall is heavy and incubators with lids are used, snow can accumulate rapidly and greatly reduce irradiance penetration into the incubator and samples. This is not common, but certainly occurs. Another challenge involves the flowing seawater system. In periods such as autumn and early

spring when atmospheric temperatures are substantially less than those of the surface water (-1.8ºC), the seawater that is normally circulating around samples to keep them at the surface temperature can freeze, usually in the incubator outflow and inflow lines, and when this happens the entire incubator can freeze quickly. Normal cautions of shading of incubators by the ship superstructure also need to be considered, given the low sun angles found for much of the photoperiod. Barber et al. (1997) found that reflection of ship structures actually increased irradiance in the incubators.

Most cruises collected samples from 100, 50, 30, 15, 5, 1 and 0.1% of surface irradiance, but others had slightly modified isolumes. The lowest isolume (0.1%) used was based on the report of El-Sayed et al. (1983) who found significant [14]C-assimilation below the 1% isolume. Given that Antarctic phytoplankton were assumed to be acclimated to low irradiance levels, the 0.1% was adopted for most cruises. For simplicity in analyses, samples from 25 and 23% of surface irradiance were pooled, as were those from 16 and 15%, 10 and 7%, and 2 and 1%.

After incubation, samples were removed and filtered through 25 mL GF/F filters under low (< ⅓ atm.) vacuum, rinsed with ca. 5 mL 0.01N HCL in cold (0ºC) seawater to remove any inorganic carbon adhering to the filter, and placed in scintillation vials (either 7 or 20 mL). Samples had an appropriate volume of liquid scintillation cocktail added, placed in the dark for at least 24 h to reduce chemiluminescence, and then counted on a liquid scintillation counter. Total added $HCO_3^-$ was determined by adding 0.1 mL unfiltered sample to a base trap (β-phenethylamine was most commonly used), and a hydrophilic

LSC fluor added and treated in a similar manner as the filters. All calculations accounted for isotope discrimination. Chlorophyll was quantified using fluorometric techniques (Knap et al., 1996) where samples were filtered through 25 mm GF/F filters, extracted in 90% acetone for 24 h in the dark at 0ºC, and the fluorescence measured before and after acidification. All fluorometers were calibrated using commercially purified chlorophyll.

     All [14]C-assimilation rates were reprocessed to insure uniform treatment. Integrated primary productivity rates were

computed through the 0.1% isolume depth, after the report of El-Sayed et al. (1983) that indicated that a 1% euphotic zone

depth was inappropriate for phytoplankton assemblages growing in relatively deep mixed layers and adapted to low light levels. The 0.1% isolume depth was set equal to zero $^{14}$C-assimilation. We note that often the 1 and 0.1% isolume samples were not statistically different (based on the number of disintegrations per minute of the filters), so that integration to the 0.1% isolume depth had little impact on integrated productivity. Assimilation numbers (carbon fixation per unit chlorophyll) were also computed for each depth. In addition to the rates of $^{14}$C-assimilation, chlorophyll and the dominant phytoplankton functional group are listed (Table 2). A total of 3,511 independent productivity values are reported.

Mixed layer depths were determined from the sigma-t values from CTD casts, and defined as a change of 0.01 kg m$^{-3}$ from a stable surface value. This is a conservative choice but was used to define a mixed layer in water columns were stratification is very weak (e.g., Smith et al., 2000, 2013). In the vast majority of stations, mixed layer depths calculated by a change of 0.01 and 0.02 kg m$^{-3}$ were the same, and a refined determination of mixed layer based on chlorophyll or fluorescence was not attempted (Carvalho et al., 2017). At the few stations where mixed layers were greater than 150 m, a mixed layer depth of 150 m was listed.

Characterization of the dominant functional group was at times qualitative. Biomass of a particular group is often difficult to directly measure, and cell numbers provide a completely different measure of the abundance and ecological importance of a particular group, as smaller cells are often much more numerically common but constitute a small portion of the total carbon-equivalent biomass. This is particularly true in the Ross Sea, where the two dominant functional groups are diatoms and haptophytes – specifically the colonial haptophyte *Phaeocystis antarctica*. Diatoms tend to be large (ranging from 10 to 200 µm), whereas individual cells of *P. antarctica* are ca. 5 µm in diameter, but are often embedded in a mucopolysaccharide sheath that houses thousands of cells (Mathot et al., 2000). Colonies reach 2 mm in diameter. Both diatoms and haptophytes have similar pigments, as both contain fucoxanthin, but *P. antarctica* having larger amounts of 19'-hexanoylfucoxanthin than diatoms; diatoms, in contrast, have chlorophyll c$_3$, which in the Ross Sea can be used with other pigments to separate the two groups (DiTullio et al., 2003). As a result, using chemical characteristics of pigments is the most powerful means of distinguishing the dominance of certain functional groups and their contribution to total chlorophyll. HPLC measurements in the Ross Sea have repeatedly shown only small amounts of phaeopigments (e.g., https://www.bco-dmo.org/dataset/3107/data), and fluorometric and HPLC chlorophyll *a* measurements are strongly correlated with a slope near 1 (Bidigare et al., unpubl.) Diatoms also have a cell wall that includes silica, and haptophytes do not; hence the presence of large amounts of biogenic silica suggest substantial accumulation of diatoms. All of these were at times used to distinguish the dominant functional group at each station. The seasonal progression of phytoplankton in the Ross Sea is generally described as an initial spring bloom of *Phaeocystis* followed by the growth and accumulation of diatoms (Smith et al., 2014), and while the description does not capture all of the spatial and temporal variability found in the Ross Sea, it is supported by the temporal pattern of dominance determined at the productivity stations.

## 3      Quality Control

As assimilation numbers (maximum chlorophyll-normalized production rates within the water column in response to natural PAR) in polar waters are in line with $P_{max}^B$ values ($P_{max}^B$ is the maximum rate of photosynthesis when normalized to chlorophyll at saturating irradiance using controlled incubations; Bouman et al., 2018) that have been measured in the Ross Sea (Smith and Donaldson, 2015), any assimilation numbers that were over four times the standard deviation of the mean assimilation number at that specific irradiance in that cruise were checked for fidelity in both the carbon assimilation rates and chlorophyll concentrations. *In situ* fluorescence patterns from the CTD casts often allowed for an assessment of the reliability of those values; if chlorophyll values were considered to be reliable, then the [14]C-uptake values were inspected for spurious values. Using this method, 6 assimilation number values were removed (from two of the 449 stations).

## 4      Results

The mean primary productivity measured by [14]C-uptake incubations was $1.10 \pm 1.20$ g C m$^{-2}$ d$^{-1}$ (Fig. 2; n = 483; minimum 10.4 mg C m$^{-2}$ d$^{-1}$ and maximum 13.1 g C m$^{-2}$ d$^{-1}$, a range of over 3 orders of magnitude; Table 3), with the minimum occurring in ice-covered, low biomass waters in April, when solar radiation was greatly reduced. The maximum rate occurred on December 22 during a large bloom of diatoms. Productivity within the water column exhibited a broad maximum from the surface to the 15% isolume, with only a modest (18.8%) decrease at the surface relative to 30% of surface irradiance (Fig. 2). Chlorophyll decreased by 13.8% at the surface relative to the chlorophyll maximum (Table 3), suggested that while photoinhibition of fluorescence and productivity did occur at the surface, on average it was relatively minor. Southern Ocean phytoplankton have a variety of responses to irradiance. One is the pigment packaging effect, which decreases the amount of light absorption per cell relative to the same absolute amount of pigment (Stuart et al., 1998). A second is variability of the accessory pigments relative to chlorophyll (e.g., Kropuenske et al., 2009), including the concentrations of xanthophyll cycling pigments which are known to be a major photoprotective mechanism. In addition, species-specific effects are known. Kropuenske et al. (2009) found that *Phaeocystis antarctica* and the diatom *Fragilariopsis cylindrus* both used xanthophyll cycling, but that the diatom had much higher rates and hence better mechanisms to cope with a transition to high irradiance. *P. antarctica* did not exhibit non-photosynthetic quenching (NPQ), while the diatoms NPQ displayed reduced quenching after exposure to high irradiance. Different species and cell sizes have a wide variety of responses to irradiance, and such strategies effectively reduce photoinhibition, but do not eliminate it. Positive productivity was often observed at the 1% irradiance depth, confirming El-Sayed's suggestion that the euphotic zone in the Ross Sea could be deeper than the generally assumed 1% light depth. Chlorophyll concentrations were also relatively uniform through the 15% light level and decreased slightly below that, although the decrease was far less than for primary productivity (Fig. 2). Assimilation numbers were also relatively uniform throughout the upper euphotic zone, although there was substantial variation in this response (Table 3). The 100, 50 and 30% isolumes averaged 0.98 mg C (mg chl)$^{-1}$ h$^{-1}$ (Fig. 2), in line with the maximum photosynthetic rate compiled for the Ross Sea continental shelf from photosynthesis/irradiance experiments (1.10 mg C (mg chl)$^{-1}$ h$^{-1}$; Smith and Donaldson, 2015).

The temporal patterns of phytoplankton composition and biomass in the Ross Sea are relatively well known (Arrigo et al.,

1999; Smith et al., 2000, 2011, 2014). Spring blooms begin at the latest by the end of October, although further analysis suggests that the onset of growth is at least one month earlier (Zhong and Smith, unpubl.). Growth in early spring is thought to be limited by irradiance, as reduced irradiance levels are imposed by ice cover, low solar angles, and deep vertical mixing. The temporal changes in mixed layer depths are similar to modelled changes and observations within one year (e.g., Smith et al., 2000; Smith and Jones, 2015), with minimum mixed layer depths occurring in mid-December and generally remaining low

through much of January (Fig. 3). Mean monthly mixed layer depths from October through February were 101, 58.5, 26.8, 21.8 and 30.6 m. Based on chlorophyll concentrations from incubations, chlorophyll was maximal around December 21, but greater than 2 µg L$^{-1}$ from Nov. 21 through the end of December (Fig. 4). Surface and integrated productivity paralleled biomass, with maxima co-occurring with chlorophyll concentrations (Fig. 4). Indeed, surface chlorophyll and surface primary productivity were significantly correlated ($R^2 = 0.669$; $p < 0.0001$) as were surface and integrated productivity ($R^2 = 0.737$; $p$

$< 0.001$). Annual integrated productivity (calculated from the data in Fig. 4) equals 146 g C m$^{-2}$ yr$^{-1}$. The maxima of all three variables correspond to the maximum of *P. antarctica* concentrations that have been repeatedly observed. However, recent investigations of temporal changes in the particulate organic carbon:chlorophyll ratios (Smith and Kaufman, 2018; Ryan-Keogh and Smith, 2021; Chen et al., 2021) suggested that summer productivity (that is, after the *Phaeocystis* bloom had dissipated) remained elevated and is not adequately assessed by chlorophyll-based satellite bio-optical models such as that of

Schine et al. (2015). The mean productivity values do not directly support this, but it is worth noting that integrated productivity rates in January were still substantial.

The importance of mixed layers in regulating the growth and accumulation of phytoplankton has long been recognized in polar oceans (Sverdrup, 1953; Mitchell and Holm-Hansen, 1990; Nelson and Smith, 1991; Smith and Jones, 2015). In general, deeper mixed layers reduce the mean irradiance available for photosynthesis, as well as mixing low chlorophyll water from

depth with waters in the surface layer which have greater concentrations of biogenic material, thus "diluting" particulate matter in the mixed layer. Mitchell and Holm-Hansen (1990) suggested that in the Southern Ocean mixed layers greater than 40 m would preclude positive photosynthesis; however, Smith and Jones (2015) showed that in the Ross Sea there were specific conditions that clearly did not support this hypothesis. In addition, it has been suggested from laboratory and field investigations that *P. antarctica* is capable of utilizing reduced irradiance levels in deep mixed layers, thus allowing it to grow

in spring (e.g., Kropuenske et al., 2009; Tozzi and Smith, 2017). Indeed, the data of Smith and Jones (2015) where deep mixed layers supported a very large standing stock of chlorophyll consisted of stations that were largely dominated by *Phaeocystis*. To see if the productivity data supported the hypothesis of Mitchell and Holm-Hansen (1990), the relationship of chlorophyll concentrations to mixed layer was analysed (Fig. 5). No simple relationship like those found in individual cruises was apparent, nor was there a relationship from stations dominated by *Phaeocystis* or those dominated by diatoms. This may have resulted

from a number of factors. Mixed layers are actually homogeneous layers, and can be substantially deeper than active mixing layers (Taylor and Ferrari, 2011). Also, the time scales of mixing can be less than one day (that is, water column mixing responds relatively rapidly to a change in wind speed or ice-generated mixing), whereas chlorophyll concentrations under low

irradiance conditions might require a number of days to respond. Determining the relevant time scales of each during an oceanographic cruise is exceptionally difficult. Finally, it is well known that phytoplankton acclimate to low irradiance conditions by increasing the amount of chlorophyll per cell, and therefore chlorophyll changes may be related more to photophysiological changes than actual growth (e.g., Geider et al., 1998). Chlorophyll also can be influenced by iron supply as well (Greene et al., 1991; Price et al., 2005). Hence the relationship of chlorophyll to mixed layer depths can be obscured by the other factors operating within the water column.

The relationship between the 1% isolume depth and integrated chlorophyll concentrations was characterized by an exponential decay response, which is expected given the importance of phytoplankton particles to irradiance attenuation. Because the data on composition is qualitative and discontinuous, a statistical analysis of the impact of assemblage composition on the relationship between chlorophyll and the 1% isolume is not possible. However, the two major groups – diatoms and *P. antarctica*- seemed to demonstrate a difference between stations dominated by haptophytes and diatoms as well as those with a mixed assemblage (Fig. 6). The equation at stations dominated by *P. antarctica* was

$$Z_{E_{1\%}} = 27.9 + 165 \times e^{-0.109 \int_{1\%}^{0} Chl} \qquad (1)$$

where $Z_{E_{1\%}}$ is the 1% isolume depth and $\int_{1\%}^{0} Chl$ is the integrated chlorophyll concentration from the surface to the 1% isolume depth ($R^2 = 0.88$, $p < 0.0001$), whereas the relationship for stations dominated by diatoms was best fit by the equation

$$Z_{E_{1\%}} = 23.4 + 28.8 \times e^{-0.0476 \int_{1\%}^{0} Chl} \qquad (2)$$

($R^2 = 0.17$, $p < 0.0001$). Such a difference needs to be assessed using controlled experiments to clarify the potential differences among the group's absorption characteristics, but it is consistent with the *in situ* data that clearly showed that diatoms have reduced carbon:chlorophyll ratios in summer under iron-limiting conditions in the Ross Sea (Smith and Kaufman, 2018; Ryan-Keogh and Smith, 2021).

Phytoplankton composition appeared to change from an assemblage largely dominated by *P. antarctica* to one largely composed of diatoms (Table 4), despite the qualitative assignment of "dominance" through the use of various metrics. The transition of an assemblage dominated by *P. antarctica* to one composed of diatoms or a mixed assemblage of diatoms and haptophytes begins in late December, and while substantial spatial variability occurs in the distribution of both groups over the entire continental shelf, it is consistent with the observations on various scales of time and space (e.g., Fragoso and Smith, 2012; Liu and Smith, 2012; Kaufman et al. 2014; Ryan-Keogh and Smith, 2021). The first date of a mixed diatom-haptophyte assemblage was on November 21, and a diatom-dominated assemblage was observed on November 26 and 27. However, most stations during November and December were haptophyte-dominated. Haptophytes never completely disappeared, but solitary cells likely became the dominant form after colonies disappeared (Smith et al., 2003; Jones and Smith, 2017). Diatoms became much more abundant and dominant in January and February. Only one station had a cryptomonad dominance; dominance by this group is likely stimulated by glacial run-off and likely is restricted to narrow bands near the coast of Victoria Land (Moline et al., 2004). Interestingly, *P. antarctica* was dominant (albeit in extremely low chlorophyll concentrations and under 100% ice cover) in April at the few stations that have been sampled. Should the species be able to remain in low concentrations

throughout winter, it might help explain its early growth as sea ice begins to recede and mixed layers shoal (and irradiance increases) in spring. Additionally, molecular analyses have suggested that *P. antarctica* may have heterotrophic capabilities (Rizkallah et al., 2020), but direct examination of this ability has so far failed to demonstrate the haptophyte's ability to grow on reduced organic substrates (D. Caron, pers. comm.).


## 5        Data Availability

All compiled data containing the 3,512 depths and 492 stations are available at the Biological and Chemical Oceanography Data Management Office (BCO-DMO) available at https://www.bco-dmo.org/dataset/863815/. Integrated water column data and PAR information are available as a supplemental file at the same site. If additional data for a specific cruise or group of

cruises is of interest, it can be obtained directly from the author and/or the requisite data repository. Data are also archived at the National Tibetan Plateau Science Data Center (http://data.tpdc.ac.cn/zh-hans/data/26d727c9-017b-46b1-be50-1a7e147d78d8/).

## 6        Recommendations for the use of these data

This data set from the Ross Sea has multiple uses and is of value to numerous investigators. The first would be as a comprehensive data base to generate new models of satellite productivity, given that present estimates of chlorophyll and biomass from satellite data are inadequate (Chen et al., 2021). These data will provide a clear data base to test the validity of productivity models not only of the Ross Sea but of other continental shelf regions in the Antarctic. A second use might be to assess the role of ecological hot spots – those restricted areas that are characterized by elevated primary productivity and serve

as regions of enhanced ecological importance in food webs. They also provide a baseline for ecological investigation of biogeochemical cycles and trophic ecology.

## 7        Conclusions

The synthesis of productivity data from the Ross Sea demonstrates the patterns of productivity that have previously only

been inferred; that is, productivity is characterized as a unimodal peak during late December, and closely follows the biomass of the system throughout the entire growing season. Productivity can be substantial; the maximum primary production rate measured was 13.1 g C $m^{-2}$ $d^{-1}$. Annual integrated productivity is estimated to be 146 g C $m^{-2}$ $yr^{-1}$. Chlorophyll-specific productivity rates in the upper surface layer average 0.98 mg C $(mg\ chl)^{-1}$ $h^{-1}$, similar to the rate found in short-term photosynthesis-irradiance experiments. Previously hypothesized relationships between mixed layer depths and productivity

are not supported by this synthesis, and no clear relationship was observed between assemblage composition and mixed layers. Photoinhibition at the surface occurred, but only decreased productivity by ca. 18%. Despite the potential uncertainties in the measurement of productivity, this synthesis confirms that the Ross Sea continental shelf is the most productive region of the Southern Ocean.

**Competing interests.** The author declares that he has no conflict of interest.

**Acknowledgements.** The author would like to thank the many co-workers, students, colleagues, and friends who helped collect the samples under sub-optimal conditions in a challenging environment. He also acknowledges the consistent funding of the

U.S. National Science Foundation (Office of Polar Programs).  Dr. Richard Barber taught him the "tools of the trade" of radioisotope work and its limitations.  Publication of this data set was funded by China NSF awards 41876228 and 41941008.

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

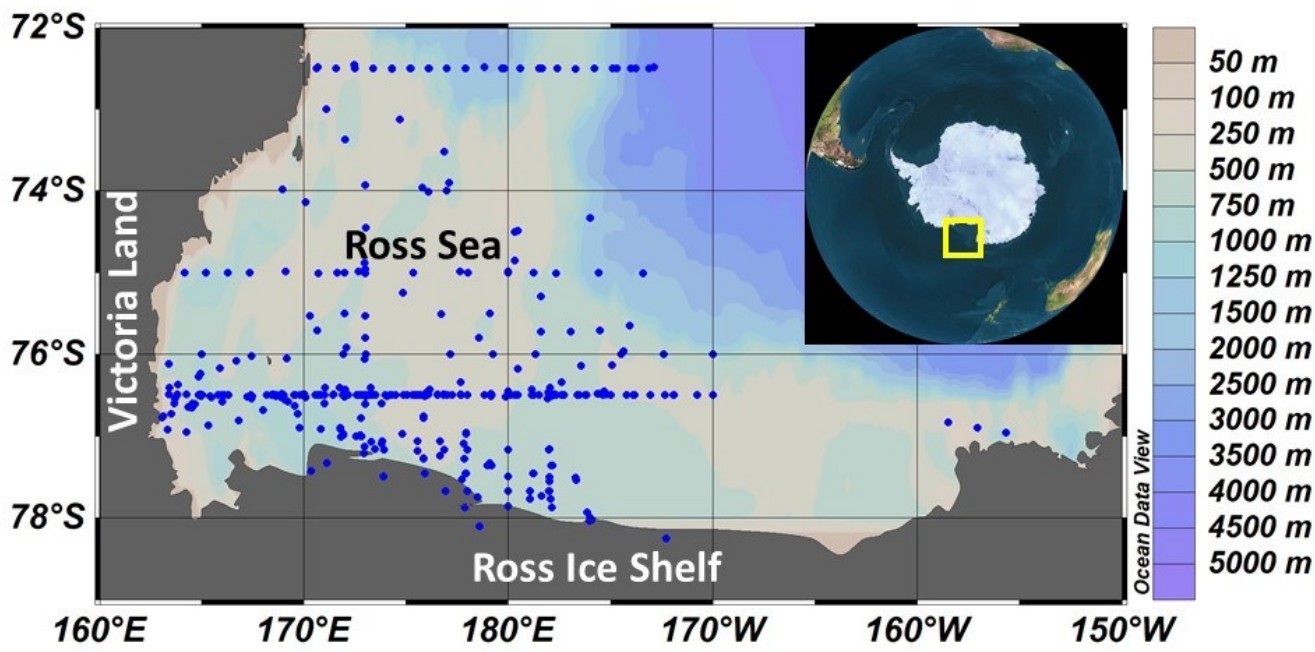


**Figure 1: Station map of all productivity stations included in this analysis. The yellow box in the insert shows the location of the Ross Sea.**

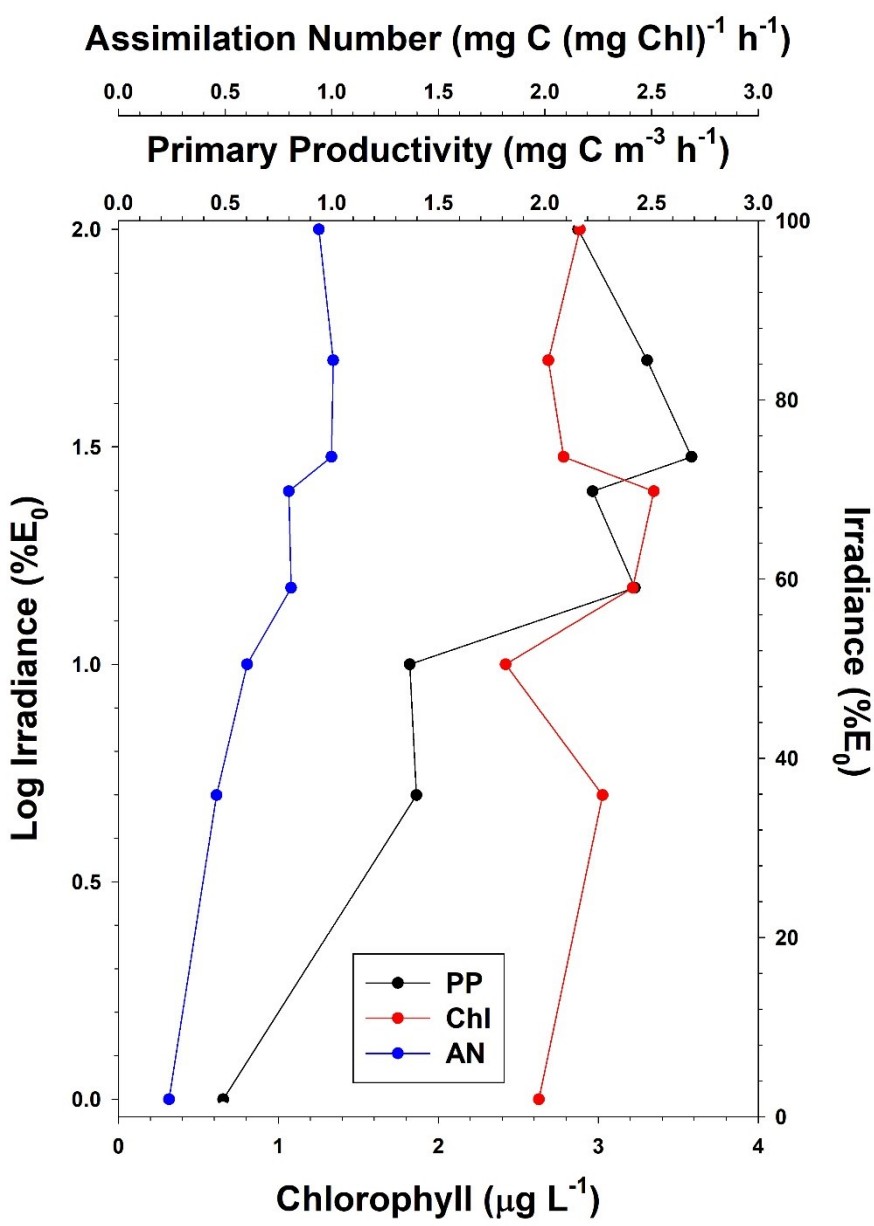


**Figure 2: Vertical distribution of average primary productivity (PP), chlorophyll concentrations (Chl), and assimilation numbers (AN) in the Ross Sea. Standard deviations for all depths and all variables are listed in Table 3.**

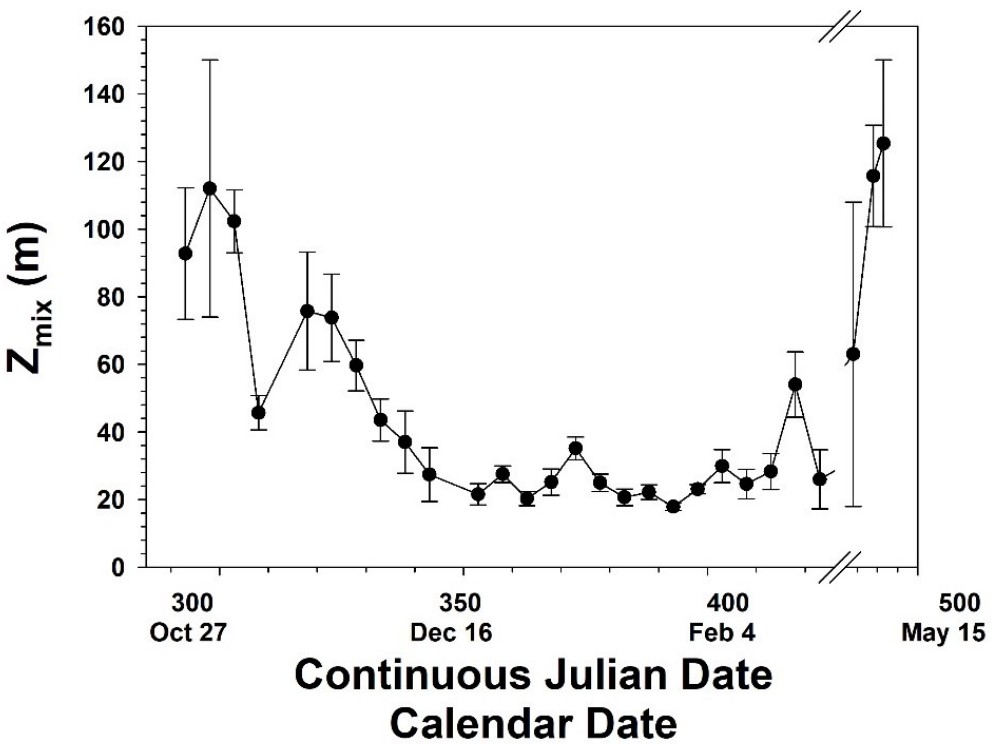


**Figure 3: Seasonal progression of mixed layer depth ($Z_{mix}$). All mixed layers greater than 150 m were set equal to 150 m. Error bars represent the standard deviation from the mean. Gaps in time were periods when no data were collected.**

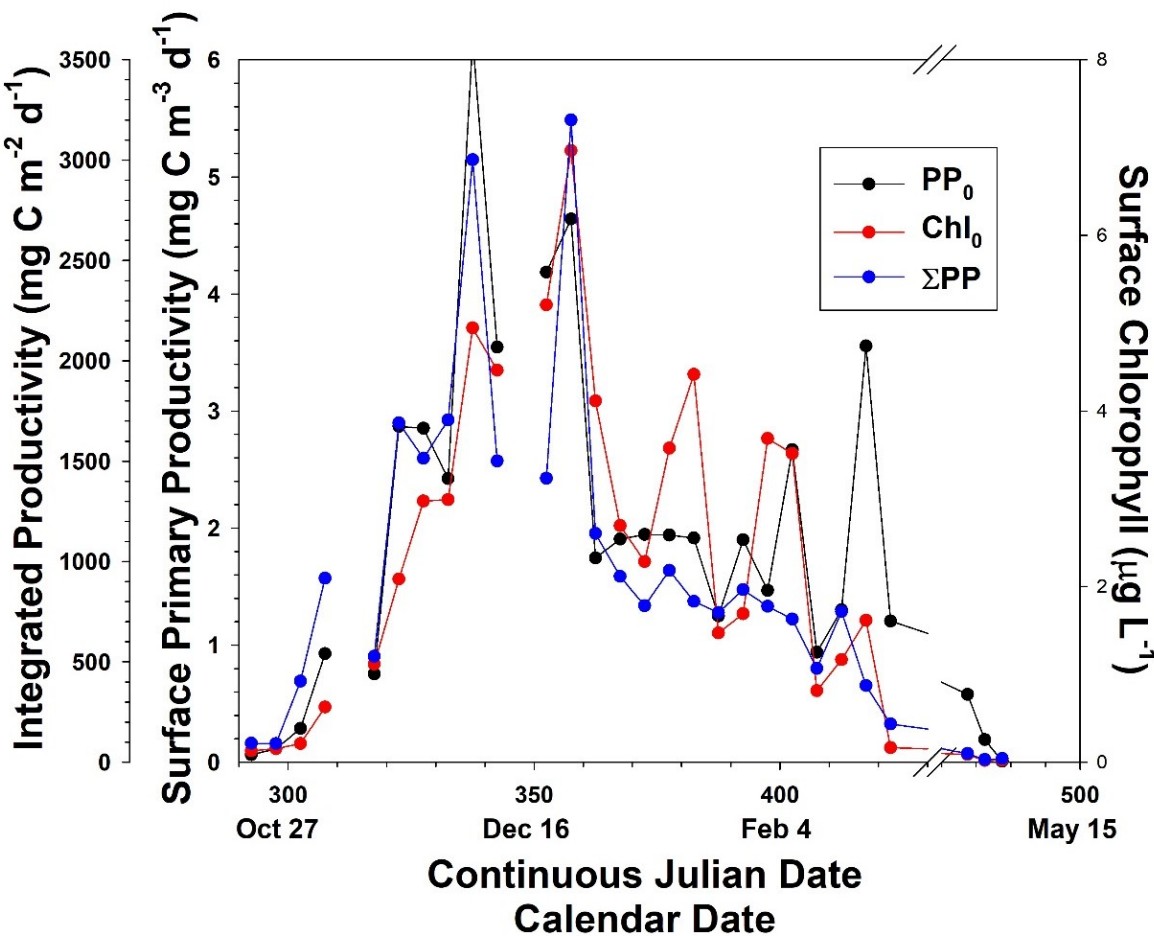


**Figure 4: Temporal distribution of surface ($PP_0$) and integrated primary productivity ($\Sigma PP$; through the 0.1% isolume) and surface chlorophyll concentrations ($Chl_0$). Data were binned in 5-day intervals, and each bin had different numbers of samples (ranging from 2 to 27 values in each bin). Gaps in time were periods when no data were collected.**





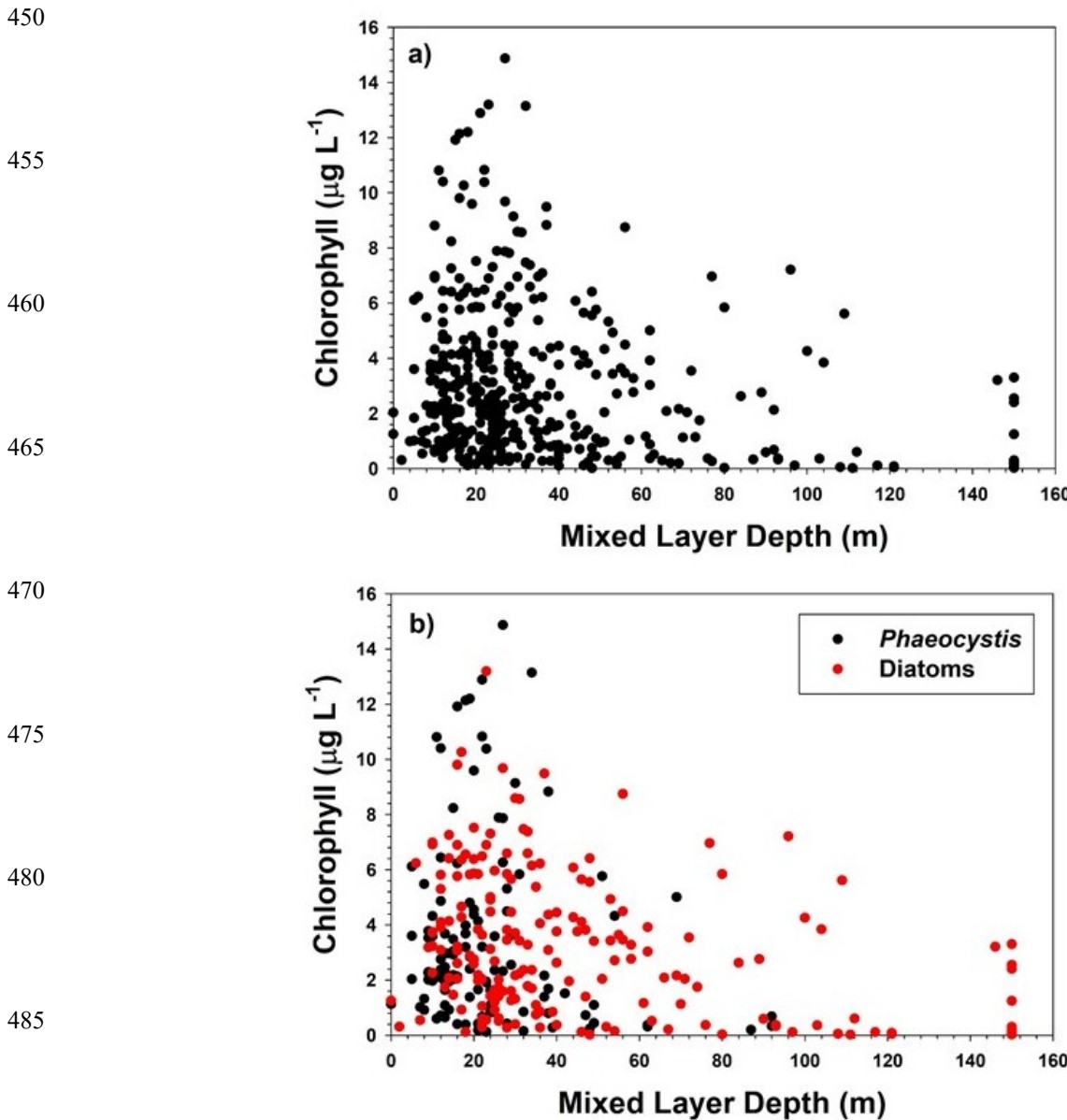






**Figure 5a) Relationship of mixed layer chlorophyll concentrations with mixed layer depth, and b) relationship of chlorophyll concentrations at stations dominated by either diatoms or *Phaeocystis* to mixed layer depths.**

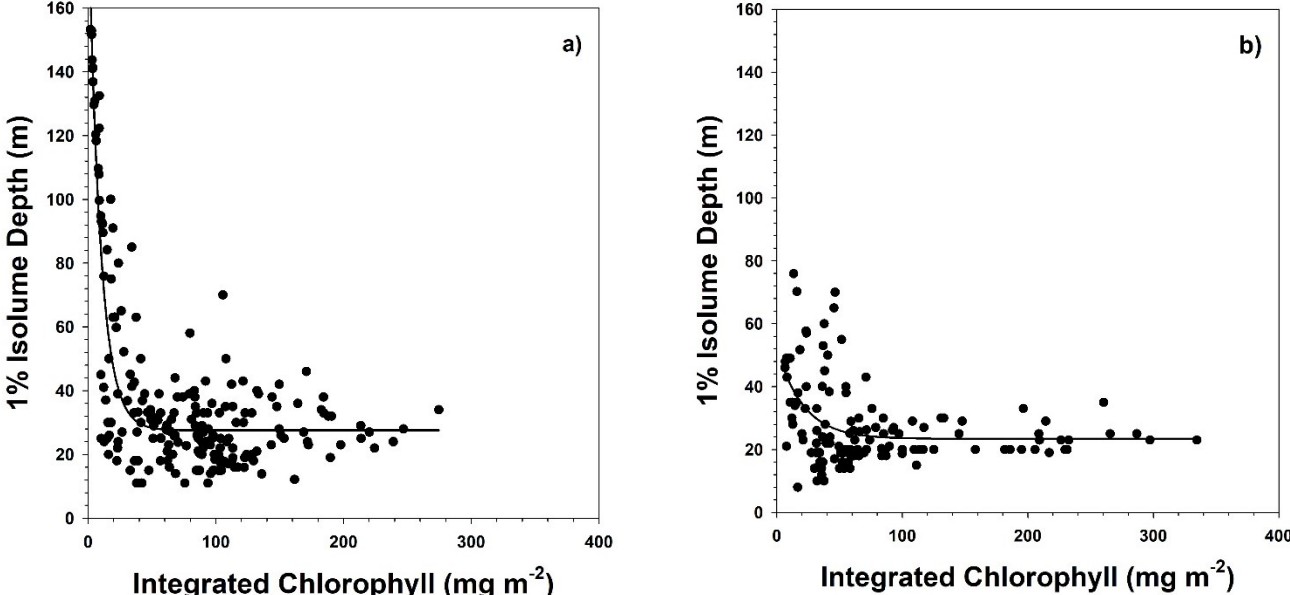

**Figure 6: Relationship between integrated chlorophyll concentrations (integrated through the 1% isolume depth) and the 1% isolume depth at a) stations dominated by *Phaeocystis antarctica* and b) stations dominated by diatoms. The relationships were best described by an exponential decay equation and were highly significant ($R^2 = 0.77$ and $R^2 = 0.17$; $p < 0.0001$ for both). Both axes were made equal to allow a direct comparison.**


**Table 1.** Name of the cruises and dates in which $^{14}$C-productivity measurements were made, along with the number of stations completed and the reference that published those data.

| Cruise | Dates | Number of Productivity Stations | Reference |
|---|---|---|---|
| *Glacier* 1983: Leg I | 1/26 – 2/2/1983 | 33 | Wilson et al. (1986) |
| *Glacier* 1983: Leg II | 2/2-8/1983 | 6 | Wilson et al. (1986) |
| *Polar Duke* 1990 | 1/13 – 2/2/1990 | 68 | Smith et al. (1996) |
| *Polar Duke* 1992 | 2/5-28/1992 | 45 | Smith et al. (1996) |
| *N.B. Palmer* 94-06 | 11/14 – 12/8/1994 | 45 | Smith and Gordon (1997) |
| *N.B. Palmer* 95-08 | 12/20/1995 – 1/20/1996 | 58 | Smith et al. (1999) |
| *N.B. Palmer 96-04* | 10/18 – 11/4/1996 | 14 | Smith et al. (2000) |
| *N.B. Palmer* 97-01 | 1/13 – 2/8/1997 | 23 | Smith et al. (2000) |
| *N.B. Palmer* 97-03 | 4/12 – 29/1997 | 12 | Smith et al. (2000) |
| *N.B. Palmer* 97-08 | 11/15 – 12/10/1997 | 34 | Smith et al. (2000) |
|  |  |  | Hiscock et al. (2001) |
| *Polar Sea* 2001: Leg I | 12/19 - 21/2001 | 8 | Smith (unpubl.) |
| *Polar Sea* 2001: Leg II | 2/2 – 6/2002 | 8 | Smith (unpubl.) |
| *Polar Sea* 2002: Leg I | 12/23 – 24/2002 | 3 | Smith (unpubl.) |
| *N.B. Palmer* 03-05 | 12-26 – 29/2003 | 9 | Smith (unpubl.) |
| *Polar Sea* 2003-2004 | 2/3 – 6/2004 | 11 | Smith (unpubl.) |
| *Polar Star* 2004 | 12/21 – 24/2004 | 11 | Smith (unpubl.) |
| *N.B. Palmer* 05-01 | 1/29 – 2/1/2005 | 13 | Smith (unpubl.) |
| *N.B. Palmer* 06-01 | 12/27/2005 – 1/9/2006 | 27 | Sedwick et al. (2011) |
| *N.B. Palmer* 06-08 | 11/20 – 12/3/2006 | 21 | Sedwick et al. (2011) |


**Table 2. Name of the columns provided in the primary productivity table along with a description of the variable and its units.**

| Header | Description | Units |
| --- | --- | --- |
| CRUISE | Name of the cruise | |
| STANUM | Station number used in publications or data source | |
| CAST/EVENT NUMBER | CTD cast number or event number assigned in cruise (if available) | |
| LAT | Latitude of sampling | Decimal degrees |
| LON | Longitude of sampling | Decimal degrees |
| DATE | Date of sample collection | Local date |
| MONTH | Month of sample collection | Local Month |
| JUL | Julian date (local) | Jan. 1 = 1 |
| JUL CONSEC | Consecutive Julian Date | Jan. 1 =366 |
| ZMIX | Mixed layer depth | m |
| INC | Length of incubation | h (rounded to nearest hour) |
| Z | Depth from which sample was collected | m |
| E | Percentage of surface irradiance that the sample was incubated | % |
| PP | Primary productivity measured by $^{14}$C-uptake | mg C m$^{-3}$ h$^{-1}$ |
| CHL | Chlorophyll concentration measured by fluorometry | mg chl m$^{-3}$ |
| AN | Assimilation number (rate of carbon fixation per unit of chlorophyll) | mg C (mg chl)$^{-1}$ h$^{-1}$ |
| INT-PP | Primary productivity integrated from the surface to the 0.1% isolume depth | mg C m$^{-2}$ d$^{-1}$ |
| INT-CHL | Chlorophyll integrated through the 1% isolume | mg chl m$^{-2}$ |
| INT-PAR | Integrated photosynthetically active radiation (400-700 nm) | mol photons m$^{-2}$ d$^{-1}$ |
| PHYTO | Dominant phytoplankton component in sampled assemblage | |
| REF | Data source or publication which describes the data most completely | |

**Table 3.** Mean and standard deviation, maximum observed, and number of measurements of primary productivity (PP), chlorophyll concentration (Chl), and assimilation number (AN) within the euphotic zone of the Ross Sea. E = percentage of surface irradiance; $E_0$ = surface irradiance; $PP_{max}$ = maximum rate of productivity at that isolume; $Chl_{max}$ = maximum chlorophyll concentration at that isolume; $AN_{max}$ = maximum assimilation number at that isolume; n = number of observations. Daily photosynthetically active radiation values for each station are available in the publicly available data set.

| E (% of $E_0$) | PP (mg C m$^{-3}$ h$^{-1}$) | $PP_{max}$ (mg C m$^{-3}$ h$^{-1}$) | Chl ($\mu$g L$^{-1}$) | $Chl_{max}$ ($\mu$g L$^{-1}$) | AN (mg C (mg chl)$^{-1}$ h$^{-1}$) | $AN_{max}$ (mg C (mg chl)$^{-1}$ h$^{-1}$) | n |
|---|---|---|---|---|---|---|---|
| 100 | 2.16 ± 2.33 | 20.2 | 2.88 ± 2.69 | 15.1 | 0.94 ± 0.71 | 4.62 | 492 |
| 50 | 2.48 ± 2.77 | 27.1 | 2.91 ± 2.69 | 19.1 | 1.01 ± 0.72 | 4.38 | 487 |
| 30 | 2.70 ± 3.40 | 25.7 | 2.80 ± 2.34 | 13.5 | 1.00 ± 0.69 | 4.09 | 319 |
| 25 | 2.75 ± 2.01 | 25.2 | 3.27 ± 3.00 | 17.1 | 0.80 ± 0.61 | 4.52 | 166 |
| 15 | 2.44 ± 2.85 | 13.9 | 3.23 ± 2.75 | 15.2 | 0.81 ± 0.63 | 4.10 | 416 |
| 10 | 1.32 ± 1.91 | 13.5 | 2.33 ± 2.50 | 10.3 | 0.61 ± 0.48 | 2.50 | 17 |
| 5 | 1.41 ± 2.39 | 31.9 | 3.04 ± 2.76 | 13.6 | 0.46 ± 0.40 | 3.36 | 472 |
| 1 | 0.49 ± 0.70 | 6.71 | 2.63 ± 2.71 | 18.8 | 0.24 ± 0.33 | 3.89 | 556 |

