# Peer review of "Primary productivity measurements in the Ross Sea, Antarctica: A regional synthesis"

_Earth System Science Data, 2021_

## Author Comment (AC3)

[revised manuscript text omitted]

---

## Author Response (AR1)

**Summary of reviewer comments and responses; all reviewer comments in italics.**

**Reviewer 1**

*The dataset provided by Dr. Walker Smith, "Primary productivity measurements in the Ross Sea, Antarctica: A regional synthesis", contains thousands of high-quality in situ measurements of chlorophyll and primary productivity in the Ross Sea, Antarctica, for a span of 23 years. Considering the importance of this region and the difficulty to obtain any field data from this remote and challenge area, this dataset is extremely valuable not only for the study of the biogeochemical processes of this region, but also for the development and evaluation of schemes for satellite remote sensing. The description of the dataset is well written. I would wish to see this dataset be quickly published and used by the oceanography and remote sensing community. I have only a few minor comments/suggestions for the author to consider.*

*1. PAR. It is the relative value (in reference to surface PAR) provided, not the absolute value. Since primary production is sensitive to solar radiation, I would like to see the inclusion of surface PAR value.*

*2. Units. The units in the dataset for chlorophyll concentration are shown as mg/L, which I think should be mg/m3, otherwise the concentrations are 1000 times larger than common values.*

*3. Format. The "-3" in "kg m-3" (line 128), the "2" in "R2" (line 215), etc., better be in superscript.*

Adding the information on PAR is a good idea, and I have done so in a revised version of the data set.  Unfortunately, not all data are accessible at this time, but will be added as soon as possible.  The data added are integrated PAR values derived from a BioSpherical sensor placed as close to the incubator as possible, and so represent the irradiance received by the samples and NOT the irradiance at the location of sampling.  All files listing these data at the BCO-DMO data site have been updated as of 2021-12-2 and will be further updated as additional archived data are accessed.

**Reviewer 2**

*Remote sensing in polar oceans is primarily challenged by its inherent issues, e.g., high Sun zenith angle. The rarity of field data is one of external problems. The data provided here is valuable for ecological research as well as in some degree for remote sensing as it covers some early periods of current operating ocean color satellites.*

*Line 85: "photoperiods were 24h, and therefore samples were collected independent of local time" is an understatement, you still cannot normalize photoperiod effects at the beginning of incubation.*

*Line 156: Did you check if those spurious values were related to any bloom development or decline?*

*Line 199: It's Fig. 5 not Fig. 4.*

*Line 211: It's Fig. 6 not Fig. 5.*

*Line 211: "No difference was apparent ...." What's the test and stats number?*

*Figure 4, variations within each bin were not displayed and discussed.*

**Response to Reviewer 2**

I agree that the data are valuable not only for bio-optical models but for ecological research in the Ross Sea. I have added a statement to that effect in the manuscript (Line 248).

Line 85: I agree that to normalize the responses to photoperiods would be difficult and even misleading, but the PAR data were requested by Reviewer 1 and added to the data files. It is of potential use in looking at effects of photoinhibition, but I did not pursue that line of investigation in this report.

Line 156: The anomalous values were carefully scrutinized prior to removal. They were confined to two stations (three depths from each station), and there may have been a duplicate inoculation of radioisotope by mistake. We don't know that for certain, but it was clear that the $^{14}$C-uptake rates were much greater than would have been expected. There was no relationship to bloom development or decline.

Line 199: Corrected.

Line 211: Corrected.

Line 211: Statistical tests and significance values now included.

Figure 4: Variations within each bin were not analyzed due to the variations in the number of data points within each bin. Those numbers are now mentioned in the figure caption. I did briefly look at the variations within the bin that had the most samples, and the variability in that bin was similar to the variability over a longer time span (e.g., weeks). A complete assessment of the within-period variability is not possible.

**Reviewer 2 – Part 2**

*Thanks for the update. My questions are addressed except the statistical test.*

*The real question is to test whether the effect of categorical factor (dominant phytoplankton group) was significant in the two-factor model (the other factor is Chla) for euphotic depth. But t-test cannot do the job, ANOVA can (http://sthda.com/english/articles/40-regression-analysis/163-regression-with-categorical-variables-dummy-coding-essentials-in-r/#example-of-data-set).*

*It's difficult to transform the nonlinear Eq1 into a linear one. Maybe we can assume 29.6 is a known constant. Then, the linear form of the regression would be:*

*Log(ZE1%-29.6) = a\*∫Chla + b\*group + c*

*Where a, b, and c are coefficients to be addressed. ANOVA will test whether the group effect is significant or not (not is expected in this case).*

**Response to Reviewer 2 – Part 2**

First, I have to admit wholeheartedly that statistics are not my area of expertise or even comfort. In addressing this comment, I had numerous discussions with a number of my colleagues here at SOO/SJTU (principally Dr. Yisen Zhong) who are far more versed in statistical methods than I. In truth, the discussions were very interesting and thought provoking to me. In summary, this is what I learned and decided to do:

1. The group data (the dominant species) is not categorical, which makes using an ANOVA problematic.
2. I then turned to an ANCOVA, but as the reviewer commented, transforming the 1% isolume data using a log transform makes things difficult or impossible, as that then impacts the group designation.

In short, I became convinced - based on advice from those that know much more about these methods than I - that there is no simple, definitive or intuitive statistic that can provide a clear separation of the effects of phytoplankton functional groups on integrated chlorophyll and irradiance attenuation (euphotic zone depth).

However, as I was intrigued by the possibility of a difference that might be reflected in these data, I replotted Figure 6 into separate panels (now Figure 6a, b). These show the best-fit power relationship

$$y = y_0 + a \times e^{-bx}$$

where y is the 1% isolume depth and x the integrated chlorophyll concentration) between integrated chlorophyll and the depth of the 1% isolume. For those stations that are dominated by *Phaeocystis antarctica,* it is a highly significant relationship (p<0.0001). I also plotted the same relationship at stations with a diatom dominance. It too was significant (but with a much lower $R^2$ value), but to me the most impressive part was that the fitted a and b values for the haptophyte stations were $165 \pm 10.0$ and $0.109 \pm 0.009$, while that at the diatom stations were $28.8 \pm 8.13$ and $0.048 \pm 0.0.0198$ (means and standard errors). To an observationalist like myself, that is a notable difference. In the manuscript, however, I simply pointed out the apparent differences between the two groups, noting that it is consistent with other observations on changing amounts of chlorophyll per cell that have been observed in the Ross Sea. I also recognize that there is a temporal component in both data sets, with diatoms largely occurring is summer and haptophytes in spring. I also completed the same analysis on the mixed assemblages, and found that their response was intermediate between the other two functional groups.

To summarize, I removed all mention of statistics, changed Figure 6 to emphasize the potential difference between functional groups, removed the original bulk fit equation and replaced it with regressions for diatoms and *P. antarctica*.

I might also add that this has stimulated additional thoughts on these relationships which are outside the scope of this paper – why both seem to approach 20 m isolume depths, the interpretation of the changes in the shapes of the curves, and more. It was an excellent learning experience, and I thank the reviewer for pushing me through this!

**Reviewer 3**

*This paper presents a valuable dataset that provides in situ measurements of the primary productivity in the Ross Sea, Antarctica which is a representative marine area in the South pole. The dataset will be very useful for understanding the carbon cycle and developing remote sensing methods of productivity estimation in Antarctic regions. The dataset is well described and analyzed. I only have a few questions for the author to consider when improving the paper.*

1. *Line 50: You mentioned that "long (e.g., 24 h) measurements approximate net production, but this has not been rigorously examined in polar systems." Have you tried to address this issue in your dataset? Does "primary productivity" in the paper mean the "net productivity" or the "gross productivity"?*
2. *According to table 1, your data collection sites in different periods are inconsistent. Could the year of measurement of these points be somehow reflected in Fig. 1? Will the distribution of data collection sites across different periods bring bias to the seasonal dynamic analysis (e.g., Fig. 4)?*
3. *Fig. 3: Why not consider using "Julian Date" for the x-axis?*
4. *Line 165: I think that photoinhibition is related to the photoprotection level of plankton which can be species-related. Therefore, I suggest mentioning the dominant phytoplankton groups and their photoprotection abilities here.*

**Responses to Reviewer 3**

1.       I never tried to thoroughly understand and investigate the "net vs. gross" issue. One of my colleagues in collecting some of these data, John Marra, looked at this issue more than anyone else, and concluded that 24-h incubations closely approximate net production. Bender et al. (2000) used multiple measures of net and gross production (oxygen isotopes, nitrate inventories) and found that in 24-h incubations "high ratios of net/gross production occurred" which suggest that net production was being approached with this incubation length. While there is undoubtedly variability in this relationship, on the whole publications on this issue strongly imply that a 24-h measurement approaches net production.
   I have now included some of the literature that has addressed this in the revised manuscript (lines 52-53).
2.       The reviewer is correct in that most of the data were collected without regard to date, and indeed, many of the studies had objectives that were linked to other oceanographic questions rather than productivity. I tried to graphically convey the temporal (both seasonal and yearly) aspects of the station locations, but concluded that it simply became "too messy" and that the overall information on sampling locations was largely adequate. I do note, and have added to the manuscript text (lines 81-85), that the seasonal differences are greater in magnitude than any observed interannual differences.

3.      I was a bit surprised at this suggestion, as Julian Date is to me less intuitive than the calendar date.  But it also is more quantitative, which is why I included it in the data set.  As a compromise, I have added Julian Date and the calendar date to both Figures 3 and 4.

4.      Again, correct.  While most studies on photoprotection have been done on natural assemblages, a number have investigated those in specific species and functional groups. In short, there are a variety of responses that can and do occur that are relevant to the Ross Sea. I have added some discussion and published literature reports on functional group responses to the manuscript (lines 176-184).

**Review: Heather Bouman**

*The Ross Sea primary productivity dataset is a valuable resource for biogeochemical and ecological modellers as well as remote sensing scientists.  It is a unique polar dataset that allows researchers to examine how changes in growth factors governs the seasonality of carbon fixation my marine algal in this productive marine environment. In addition to making this dataset freely available the author also provides an expert narrative of the ecology and biogeochemistry of the region.*

*P1 L18 – photosynthesis/irradiance measurements, I would instead write as photosynthesis-irradiance experiments.*

*General comment: although Phaeocystis antarctica is italicised in the abstract, in the main text it is not (nor is P. antarctica).  Also for units, exponents are not superscript on P5, some instances on P6.*

*P3 L76 – I would also add ocean biogeochemical models.*

*P3 Could the author provide more details on the type of plastic bottles used (typical volume? polycarbonate?) and the source of the blue filters used?*

*P 4  The author mentioned that HPLC phytoplankton pigments were measured alongside fluorometric chlorophyll.  Was the relationship robust between the two across the seasons/regions sampled?  Was there any evidence of the presence of phaeopigments?*

*P5 "diatoms, in contrast, have chlorophyll c3". The work of Wright and others suggests that c3 is more widespread in the haptophytes (including coccolithophores and Phaeocystis) and only found in some species of diatoms (e.g. Pseudonitzschia)?  Is the author suggesting that chl c3 is a marker of diatom presence?*

*P5 L147 Phaoecystis – typo*

*P 5 151  is the maximum rate of photosynthesis at saturating light levels using controlled light incubations (in the absence of photoinhibition).  This should not be confused with chlorophyll-normalised rates in situ (e.g. ) which uses solar light that changes (diel and diurnal variability).*

*Was the same type of blue light filter used for all bottles, in terms of its spectral distribution? Is the author stating that the chlorophyll-normalised values from the deck incubations were in line with those determined using conventional P-E incubations? This could be clarified.*

*P5 L159-160 On the pdf copy the values reported for the daily uptake rates have a comma rather than a decimal point, so look like they are a factor of 10³ too high? I suspect that this is likely a typesetting issue.*

*Fig 2 & Table 3. The standard deviations are reported in Table 3 and standard errors are reported in Figure 2. Since the dataset is provided the sample statistics can be directly computed, perhaps it is sufficient to just show the profiles, in which case I would include the standard deviations to show that despite the cold temperatures and corresponding low assimilation numbers, there is still considerable variability in the chlorophyll-normalised rates of carbon fixation.*

**Responses to Heather Bouman**

Thank you for your comments, especially since the paper was modeled after Bouman et al. (2018) in ESSD! Most are easy to respond to, and below is the listing of changes I have made.
1. L. 18: "/" changed to "-"
2. General comment re italics: In the version I am looking at *Phaeocystis* and *P. antarctica* are all italicized. I can only imagine that during pdf formation those were removed, but prior to submission of the revision, I will check those carefully. Similarly, all exponents were checked and all were correct in the version I have.
3. "Biogeochemical models" was added (now L. 77).
4. Information is now provided on the types of bottles used as well as the blue filters used (L. 90-91).
5. HPLC pigment data are not available for all cruises, but the various data sets referenced present HPLC data when available. A statement has been included (L. 152-154) that states that while phaeopigments are present, they are in very low concentrations (suggesting that microzooplankton grazing is minimal, although this is not mentioned in the manuscript), and that fluorometric and HPLC chlorophyll *a* values are highly correlated with a slope near 1.
6. Heather is correct in that chlorophyll $c_3$ is not simply confined to diatoms. However, in the Ross Sea its presence is largely confined to diatoms that occur there (coccolithophorids do not occur in the Ross Sea), making it a reliable pigment to separate functional groups, especially when used with other HPLC pigments. The statement has been modified (L. 149) to reflect this more accurately.
7. The statement about the comparison of assimilation numbers and $P^B_{max}$ has been hopefully clarified (L. 161-163).
8. A revised statement concerning the blue filters (L. 90-91) now indicates that the same filters were used for all productivity determinations as used in controlled P-E incubations.
9. Sadly, this was not a typesetting issue, but a terrible error on my part. It has now been corrected.
10. The standard deviations were not plotted initially because they were large, and the plot became a bit unmanageable and the vertical trends obscured. Heather's comment is correct; the standard deviations were listed in Table 3. In this revision the standard errors in Figure 2

are removed and only the means plotted, retaining the clarity for the vertical information. The standard deviations are retained in Table 3 but are also mentioned in the figure caption for Figure 2 as well. An additional comment is included in the text (L. 188-189) that emphasizes the large variability in the chlorophyll-normalized rates of carbon fixation.

---

## Author Response (AR2)

Responses to Editor comments

1.  In Table 3 the column listing the ln of irradiance was removed as per suggestion. Absolute irradiance levels were not listed, since taking a mean irradiance value over all stations doesn't illustrate much other than showing the variability among all stations.  In addition, the number of integrated PAR readings is fewer than the number of productivity estimates, which then leads to a comparison of different populations of data.  I did note in the table caption that absolute irradiance is available in the data file.

2.  The data have been submitted to the second data base as suggested.